# Radiological Hazard Evaluation of Some Egyptian Magmatic Rocks Used as Ornamental Stone: Petrography and Natural Radioactivity

**DOI:** 10.3390/ma14237290

**Published:** 2021-11-28

**Authors:** El Saeed R. Lasheen, Mohammed A. Rashwan, Hamid Osman, Sultan Alamri, Mayeen U. Khandaker, Mohamed Y. Hanfi

**Affiliations:** 1Geology Department, Faculty of Science, Al-Azhar University, Cairo P.O. Box 11884, Egypt; elsaeedlasheen@azhar.edu.eg; 2Geological Sciences Department, National Research Centre, 33 El Bohooth St. (Former El Tahrir St.), Dokki, Giza P.O. Box 12622, Egypt; ma.attia@nrc.sci.eg; 3Department of Radiological Sciences, College of Applied Medical Sciences, Taif University, Taif 21944, Saudi Arabia; ha.osman@tu.edu.sa (H.O.); s.alamri@tu.edu.sa (S.A.); 4Centre for Applied Physics and Radiation Technologies, School of Engineering and Technology, Sunway University, Bandar Sunway 47500, Malaysia; mayeenk@sunway.edu.my; 5Institute of Physics and Technology, Ural Federal University, St. Mira, 19, 620002 Yekaterinburg, Russia; 6Nuclear Materials Authority, Maadi, Cairo P.O. Box 530, Egypt

**Keywords:** ornamental stone, gamma-ray spectrometer, radiological hazard indices, natural radioactivity

## Abstract

Magmatic rocks represent one of the most significant rocks due to their abundance, durability and appearance; they can be used as ornamental stones in the construction of dwellings. The current study is concerned with the detailed petrography and natural radioactivity of seven magmatic rocks. All are commercial granitic rocks and are identified as black Aswan, Nero Aswan, white Halayeb, Karnak, Verdi, red Hurghada and red Aswan. Their respective mineralogical compositions are classified as porpheritic granodiorite, granodiorite, tonalite, monzogranite, syenogranite, monzogranite and syenogranite. A total of nineteen samples were prepared from these seven rock types in order to assess their suitability as ornamental stones. Concentrations of ^226^Ra, ^232^Th and ^40^K radionuclides were measured using NaI (Tl) scintillation gamma-ray spectrometry. Among the studied magmatic rocks, white Halayeb had the lowest average values of ^226^Ra (15.7 Bq/kg), ^232^Th (4.71 Bq/kg) and ^40^K (~292 Bq/kg), all below the UNSCEAR reported average world values or recommended reference limits. In contrast, the other granitic rocks have higher values than the recommended limit. Except for the absorbed dose rate, other radiological hazard parameters including radium equivalent activity, annual effective dose equivalent, external, and internal hazard indices reflect that the White Halyeb rocks are favorable for use as ornamental stone in the construction of luxurious and high-demand residential buildings.

## 1. Introduction

Ornamental stone represents one of the dominant industrial economies worldwide, and its demand shows geometric growth with the increasing construction of luxurious dwellings [1,2,3]. Egyptian basement rock constitutes the northwestern sector of the Arabian Nubian Shield (ANS), which crops out in the Eastern Desert, south Sinai and southwestern side of the Western Desert [4]. In the Eastern Desert, these rocks extend from the north, near Cairo, to the south along the Sudanese border. Granitic rocks represent the main magmatic rocks, making up ~60% of the total Egyptian basement rock of the Nubian Shield [4]. Granitic rocks represent one of the most important ornamental stones, along with marble [4]. The former is characterized by its durability and prestigious shape, and can be used as decorative stone in floors, stairs, walls, bridges, and sculptures [5,6]. Egyptian quarries represent one of the main producers globally as one of the top eight raw material-producing countries, producing about 3.2 million tons of quarried stone under twenty-five different brands. Moreover, Egypt represents the seventh-largest ornamental stone exporting country globally, amounting to 1.5 million tons/year [7].

The distribution of uranium in the Earth’s crust is associated with magmatic activity through the creation of the Earth. Depending on the amount of natural radionuclides, the radioactivity concentration in soil is relatively higher in volcanic, phosphatic, granitic and salty rocks. These rocks break into pieces over time under environmental conditions and spread across the environment. Uranium, thorium and potassium concentrations are mostly higher in the granitic rocks, granitic pegmatites and syenites, and are closely linked to mineralogical composition and petrographic features [8,9,10,11]. In general, felsic igneous rocks have greater levels than sedimentary rocks. Primary uranium minerals such as uraninite, pitchblende and coffinite are formed during rock formation, while secondary ones (e.g., uranophane) are formed later due to hydrothermal solution [12].

Enrichment of some elements such as Rb, Nb, Ta, U, Th, Zr, K and REEs is related to protracted fractional crystallization of the magma toward felsic rocks. Therefore, the previous elements’ abundance increases with increasing alkalinity (K_2_O + Na_2_O) [9]. In addition, felsic magmatic rocks require a high SiO_2_ content; thus, felsic rocks can be viewed as crust-derived rocks. In addition, zircon, allanite, titanite, xenotime, apatite, monazite and thorite minerals are the main accessory minerals in felsic rocks, including granitic rocks, as U and Th are more easily able to enter the lattice structures of these minerals; they thus represent a U and Th reservoir [11]. On the other hand, enrichment of U and Th in felsic rocks may be ascribed to magmatic segregation and hydrothermal activities, which can lead to different alteration processes [9,13].

Radioactivity can be found in rocks, soil, beach sand, sediment, river bed, rivers and oceans, and even building materials and dwellings. Naturally occurring radioactive materials usually have a terrestrial origin (primordial radionuclides), left behind since the earth’s creation [14]. They are typically long-lived, with half-lives ranging from hundreds of millions of years to billions of years. Natural sources of gamma radiation (background radiation) are predominantly attributable to primordial radionuclides, primarily the ^232^Th and ^238^U series and their decay products, as well as ^40^K, all of which exist at trace amounts in the earth’s crust. The natural radiation of soil and rocks depends on their mineralogical composition [15,16]. Measuring the concentration of radionuclides helps in monitoring environmental radioactivity. Radionuclide distributions vary from one rock to another depending on rock type. Radiological impacts on human health can be inferred by detecting the distributions and concentrations of natural radionuclides in those rocks/soil [17,18] generally used as raw materials in the construction of buildings and infrastructure.

The examined rocks were prepared as cubes in order to assess their suitability as ornamental stones through both physical (water absorption, bulk specific gravity, actual density) and mechanical (compressive strength, abrasion resistance) tests (to be reported elsewhere). Here, the current study aims to identify some magmatic rocks’ mineralogical constituents and determine the concentrations of natural radionuclides (^238^U, ^226^Ra, ^232^Th and ^40^K). The measured data may help to assess the concomitant radiological hazards to human health resulting from exposure to emitted gamma radiation from those studied granitic rocks that show the greatest application for the construction of dwellings and infrastructure.

## 2. Materials and Methods

### 2.1. Radiation Shielding Capacity

The Egyptian Neoproteozoic rocks used in the present study are widely distributed along the Red Sea coast (Eastern Desert) as well as the Sinai. Particularly, granitoid rocks vary from older/syn-tectonic (OG, grey granites) to younger/late to post-tectonic granites (YG, pink granites) [19,20]. The Arabian Nubian Shield represents the northern zone of East Africa orogen, formed during the closing of the Mozambique Ocean and the collision of East and West Gondwana [9,21,22]. Although granitic rocks occur in the Oweinat area, the relative abundance of younger granites to older granites increases from 1:4 in the south to 1:1 in the north of the Eastern Desert and 12:1 in the Sinai [23] (Figure 1). Different commercial granitic rocks were collected from various distributed localities (seven quarries; Figure 1).

The examined popular classes Nero Aswan, black Aswan, red Aswan, red Hurghada, yellow Verdi, white Halayeb and Karnak were collected from quarries (Figure 2). All samples were polished and prepared as equidimensional cubes to assess their suitability as ornamental stone, then girded to measure their natural radioactive concentrations.

### 2.2. Petrographic Investigation

The detailed petrographic investigation was carried out using point-counting techniques by polarizing microscope (Olympus bx53) for popular granitic rocks that were already used as an ornamental stone in order to detect their mineralogical composition textural relationships. Based on modal analysis of mineralogical composition, the examined magmatic rocks, including black Aswan, Nero Aswan, white Halayeb, Karnak, Verdi, red Hurghada and red Aswan, were classified as porpheritic granodiorite, granodiorite tonalite, monzogranite, syenogranite, monzogranite and syenogranite, respectively.

Black Aswan (1) was classified as porphyritic granodiorite. It has a medium to coarse grain and is composed essentially of plagioclase, quartz, potash feldspars, hornblende, biotite and a subordinate amount of augite. Plagioclase (58 vol.%) occurs as tabular, subhedral crystals that exhibit slight to extensive alteration (Figure 3a). Potash feldspars (14 vol.%) are represented by microcline and perthite. Microcline is common as fine-grained and subhedral crystals mostly corroded by quartz. Rapakivi textures are abundant. Quartz (22 vol.%) occurs as an anhedral crystal, fine- to medium-grained, that reveals undulose extinction. Hornblende occurs as a medium to coarse grained anhedral to subhedral crystal, partially altered to chlorite. Biotite occurs as fine to medium grained irregular flakes slightly altered to chlorite and muscovite, especially on cleavage planes and their periphery. Augite rarely occurs as anhedral crystals, high relief, and perfect perpendicular cleavage. Titanite, apatite and iron oxides as accessory minerals have also occurred (Figure 3b).

Nero Aswan (2) was classified as granodiorite, which is composed mainly of plagioclase (55 vol.%), quartz (19 vol.%), and potash feldspars (15 vol.%), with subordinate hornblende and biotite based on their modal analysis. K-feldspar is represented by orthoclase perthite that is partially altered to sericite (Figure 3c). Titanite, allanite and zircon are present as accessory minerals. This is similar to black Aswan; however, the mineralogical constituents are slightly altered, certainly plagioclase and K-feldspars. Moreover, allanite and zircon are present accessory minerals (Figure 3d).

White Halayeb (6) was classified as tonalite. It is deformed, fine to medium grained and composed mainly of plagioclase, quartz, biotite, and a minor amount of muscovite. Epidote, chlorite and saussurite are the main secondary minerals. Quartz (16 vol.%) occurs as fine to coarse grains and reveals undulous extension, fractured and occasionally filled by sericite due to deformation processes (Figure 3e). Plagioclase (60.4 vol.%) is the most dominant mineral and is tabular with pericline, zoned, and lamellar twinning, altered to saussurite and epidote (Figure 3f). The main secondary mineral is epidote, which occurs as a well-developed subrounded mineral with high interference colour ranging from yellow to blue, scattered on the periphery of plagioclase as an alteration product.

Red Aswan (3) was classified as syenogranite, with megacrystals of K-feldspar, plagioclase and quartz. K-feldspar (36 vol.%) is represented by pristine, patchy and very coarse-grained microcline crystals (Figure 3i). Plagioclase (~20 vol.%) is subhedral and tabular, and partially or completely altered to saussurite, kaolinite and epidote. Normal, rarely-fractured quartz is present. Short zircon crystals occur as a euhedral, high-relief and embedded in biotite. Allanite occurs as a coarse-grained crystal and represents an alteration product of biotite (Figure 3j).

Hurghada (4) granitic rocks were classified as monzogranite. They are similar to Verdi in terms of their turbid, dusty surface, K-feldspar, and grain size. Coarse-grained and turbid surfaces represent potash Feldspars. Quartz occurs as a medium to coarse grained anhedral crystal and exhibits undulose extinction (Figure 3l). Occasionally, it is fractured and filled by secondary sericite. Most of the plagioclase crystals are kaolinitized and saussuritized and reveal lamellar and pericline twinning. In addition, biotite occurs as fine-grained flaky crystals, and is partially altered to chlorite and stained by iron oxides.

Verdi (5) was classified as syenogranite, with K-feldspar (~45 vol.%), plagioclase (~17 vol.%), and quartz (~33 vol.%) with normal extinction. K-feldspar occurs as coarse-grained, turbid (kaolinitized) microcline perthite (Figure 3k). Plagioclase mostly occurs as slightly kaolinitized tabular crystals. Titanite is the main accessory mineral, with a well-developed sphenoidal shape.

Karnak (7) granitic rocks were classified as monzogranite, consisting mainly of K-feldspars, quartz, plagioclase and biotite. K-feldspars (27 vol.%) are represented by pristine microcline (Figure 3g), medium to coarse grained and sometimes enclosing fine-grained saussuritized plagioclase. Quartz (31 vol.%) occurs as a medium to coarse grained subrounded crystal with normal extinction. Plagioclase (~35 vol.%) is represented by subhedral tabular crystals, and sometimes engulfs other constituents such as biotite and quartz (Figure 3h). Flaky biotite crystals are twisted and transformed into chlorite and iron oxides.

### 2.3. Analytical Techniques

Nineteen samples were collected from seven quarries: Nero Aswan (three samples), black Aswan (three samples), red Aswan (three samples), red Hurghada (three samples), Verdi (two samples), white Halayeb (three samples), and Karnak (two samples). Identification and nomenclature of the examined rocks were carried out using point-counting techniques by polarizing microscope (Olympus bx53), according to the relative amounts of potash feldspar, quartz and plagioclase. These samples were crushed and ground, and 350 gm from the ground samples were packed in a plastic container and sealed for about thirty days to attain secular equilibrium between parents and short-lived progeny. An NaI(Tl) scintillation gamma-ray spectrometer with a crystal size of 76 mm × 76 mm was used to determine the concentrations of radioelements (^226^Ra, ^232^Th and ^40^K) in the granitic samples. A low background measurement environment was ensured by placing the detector in an arrangement that was enclosed in a cylindrical lead shield with a diameter of 15.7 cm, a length of 20.5 cm, and a thickness of 3.7 cm, with an attenuation factor of 0.16 (stopping at around 84% of input photons) for 2.6 MeV gamma rays. A spectroscopic amplifier and a multi-channel analyzer were part of the pulse processing and data analysis system, which was linked to an IBM-compatible computer. The corresponding gamma energies of ^226^Ra, ^232^Th and ^40^K were 1764 keV (Iγ = 15.30%) from ^214^Bi, 2614 keV (Iγ = 99.754%) from ^228^Ac, and 1460 keV (Iγ = 10.66%), respectively [24,25]. Approved reference materials, such as RGU-1, RGTh-1, and RGK-1 were used, and their densities after pulverization were close to those of the building materials [26]. The design of the container was chosen based on the premise that the radioactivity in the measuring samples was evenly distributed. The samples were counted for 2000 s, with MDAs of 2, 4, and 12 Bq kg^−1^ for ^226^Ra, ^232^Th, and ^40^K, respectively. The overall uncertainty of the radiation levels was calculated using the error propagation law of systematic and random measurement errors. Systematic errors of 0.5 to 2% existed in the efficiency calibration, and random errors of up to 5% existed in the radioactivity readings [27].

## 3. Results and Discussion

### 3.1. Radionuclides Concentrations

Radionuclides (^226^Ra, ^232^Th and ^40^K) were measured by multichannel gamma-ray spectrometer, and are listed in Table 1. White Halayeb had the lowest average values of ^226^Ra (15.7 Bq/kg), ^232^Th (4.71 Bq/kg) and ^40^K (~292 Bq/kg). In addition, it possessed radionuclide values lower than the recommended average worldwide values (32 Bq/kg for ^226^Ra, 45 Bq/kg for ^232^Th, and 412 Bq/kg for ^40^K) according to [13] and [28]. Controversially, it is noticeable that the other studied samples had average radionuclide values higher than the average worldwide values. This may be ascribed to incorporation of radionuclides in the crystal structure of some accessory minerals, such as zircon, sphene and allanite [29]. On the other hand, the greater radium concentration is due to the modification of radioactive materials deposited within granite fractures. Furthermore, the leaching of uranium minerals (uranophane, uraninite, and betauranophane) through rainwater aids in their migration and precipitation along joints and faults [30].

Assuming equilibrium between uranium and radium, the mean of the ^232^Th/^226^Ra ratio in the examined samples of magmatic rocks was less than the mean worldwide value of crustal rocks, 3.94 [15]. The lowest results contributed to the high Ra content relative to Th in the studied samples, reflecting Ra mobility relative to Th, especially in the secondary environment. This finding is linked to the slightly elevated uranium activity in the study area; radium content is derived from uranium-bearing granitic rocks [20].

Activity concentrations of ^226^Ra, ^232^Th, and ^40^K in the examined granites were compared with others from Egypt and different countries (Table 2). The obtained results of radionuclide activity concentrations are observed to be higher than the values in Saudi Arabia and Oman, and lower than the other regions. The comparison shows how radioactivity varies from one country to the next, which is linked to differences in the geological structures of constituent rocks in each country.

### 3.2. Radiation Hazard Assessment

The radiological hazard of the examined samples can be estimated using the obtained values of radionuclide (^226^Ra, ^232^Th and ^40^K) concentration in order to determine their safe utilization as a decorative stone. The assessment indices include absorbed dose rate (D), annual effective dose (AED), radium equivalent activity (Ra_eq_), and external ((H_ex_) and internal (H_in_) hazards.

#### 3.2.1. Absorbed Dose Rate (D)

The rate of absorbed gamma dose can be calculated for the distribution of the unique radionuclide in the air at 1 m above the ground surface [42,43]:D (nGy/h) = 0.462A_Ra_ + 0.604A_Th_ + 0.0417A_K_(1)
where A_Ra_, A_Th_ and A_K_ are the specific activity concentrations of ^226^Ra, ^232^Th and ^40^K in Bq/kg, respectively.

Among the calculated absorbed gamma dose rate, only the white Halayeb sample recorded the lowest value of 22.04 nGy/h (Figure 4), which lies within the safety limit (59 nGy/h) [42,44]. This shows that all granitic rocks except white Halayeb are unsuitable for infrastructure, especially building materials.

#### 3.2.2. Annual Effective Dose (AED)

The annual effective dose (AED) was calculated from the absorbed dose by applying the dose conversion factor of 0.7 Sv/Gy and the outdoor occupancy factor of 0.2 [40]:AED (mSv/y) = D(nGy/h) × 8760(h) × 0.2 × 0.7 (Sv/Gy) × 10^−6^(2)

The average value of AED obtained from the white Halayeb samples was lower than the recommended limit (0.07 mSv y^−1^, ref [39] (Figure 4), while the average of AED for the rest of granitic rocks exceeded the recommended limit. Heavy minerals found in granites, such as monazite, uraninite, and thorianite, can be responsible for high radiation exposure. Furthermore, long-term exposure to gamma radiation may have such negative health consequences as tissue degeneration, deoxyribonucleic acid (DNA) in genes, cancer, and cardiovascular disease. Refs [45,46,47].

#### 3.2.3. Radium Equivalent Activity (Ra_eq_)

Activity levels of ^226^Ra, ^232^Th and ^40^K in the examined samples can be determined using the radium equivalent activity (Ra_eq_) index. This index can be calculated using equation (3) [40]:Ra_eq_ (Bq/kg) = A_Ra_ + 1.43A_Th_ + 0.077A_K_(3)

The average of the obtained values ranges from 36.10 Bq/kg in white Halayeb to 418 Bq/kg in red Hurghada. Furthermore, the constraints for all the calculated values are less than the maximum permissible value of 370 Bq/kg; [27], suggesting a negligible radium equivalent impact for the rocks (Figure 4). On the other hand, the highest values detected in the red Hurghada samples mean that it is not safe to utilize as an ornamental stone and building material.

#### 3.2.4. External Hazard Index (H_ex_)

In order to assess the rate of radiation dose emitted from natural radionuclides in the examined samples, an external hazard index (H_ex_) was applied using the following equation [46]:H_ex_ = A_Ra_/370 + A_Th_/259 + A_K_/4810(4)

The mean calculated external hazard index ranged from 0.12 in white Halayeb to 0.83 in red Hurghada, which is less than unity and within the safety limit, reflecting a negligible radiation hazard (Figure 4).

#### 3.2.5. Internal Hazard Index (H_in_)

Exposure to radon (^222^Rn) and its radioactive daughters is detected by the internal hazard index (H_in_), which can be used to measure impact on the respiratory organs and lungs [46]:H_in_ = A_Ra_/185 + A_Th_/259 + A_K_/4810(5)

All of the examined rocks had average internal hazard values lower than the limiting value of 1, except red Hurghada which had a value of 1.1. This reflects that the internal hazard values lie within the safety limit (Figure 4). Radiation hazard indices H_ex_ and H_in_, with higher values, suggest a significant risk to human health. When granite is used as a construction material, external gamma rays and radon gas inhalation do not cause any radioactive health risks [42].

The present study discusses the link between petrographic studies and natural radioactivity. It is noticeable that there are strong associations between petrography (identifying the mineralogical constituents and textural relationships) and natural radioactivity. For example, White Halyeb is classified as tonalite, a type of older granite, and represents the oldest unit in this study. It consists mainly of plagioclase, quartz, biotite, and a minor amount of muscovite. A lack of accessory minerals (e.g., Allanite, monazite, titanite, thorite, and zircon) yields a low content of activity concentrations of natural radioactivity. Therefore, White Halyeb is more suitable as a decorative ornamental stone.

## 4. Conclusions

Nineteen samples of different rocks representing seven magmatic rocks in named commercial granitic classes were examined for their mineralogical, petrographic and radioactive compositions in this study. Mean activity concentrations of ^226^Ra, ^232^Th, and ^40^K in all of the examined rocks except the White Halyeb exceeded the limiting range suggested by the UNSCEAR. Most of the calculated radiological hazard parameters suggest that White Halyeb is the most suitable rock for use as an ornamental stone or building materials, as it possesses a low level of natural radioactivity, and hence poses little or no radiation risk to human health.

## Figures and Tables

**Figure 1 materials-14-07290-f001:**
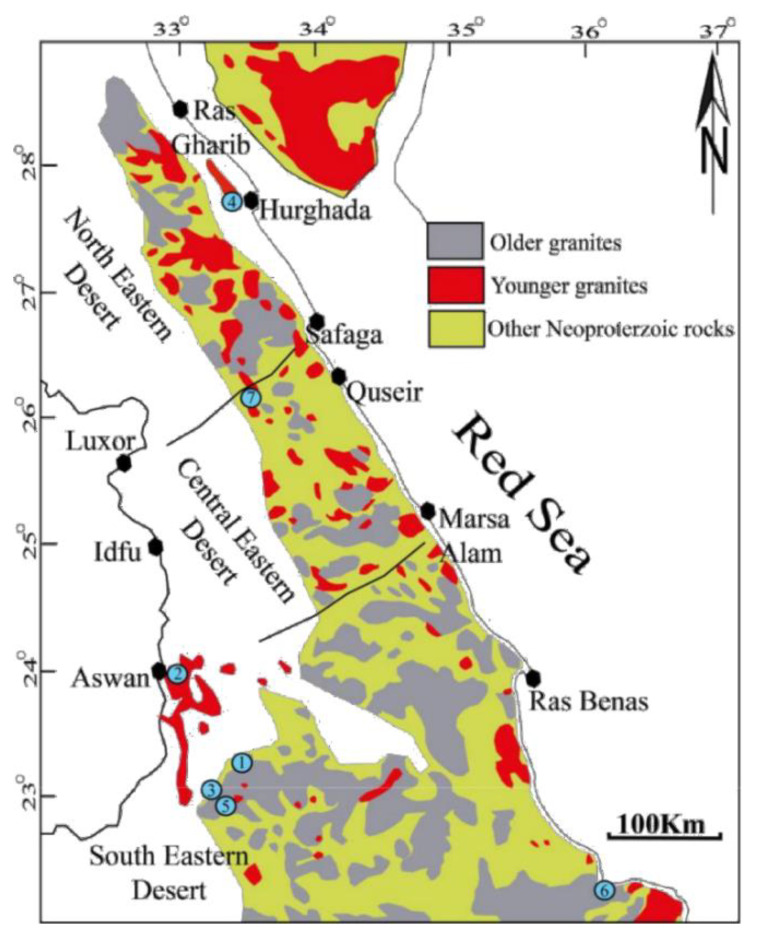
Distribution map of Neoproterozoic rocks in the Eastern Desert and Sinai after [10], including sample locations over a wide area: (**1**) black Aswan, (**2**) red Aswan, (**3**) Nero Aswan, (**4**) red Hurghada, (**5**) yellow Verdi, (**6**) white Halayeb, (**7**) Karnak.

**Figure 2 materials-14-07290-f002:**
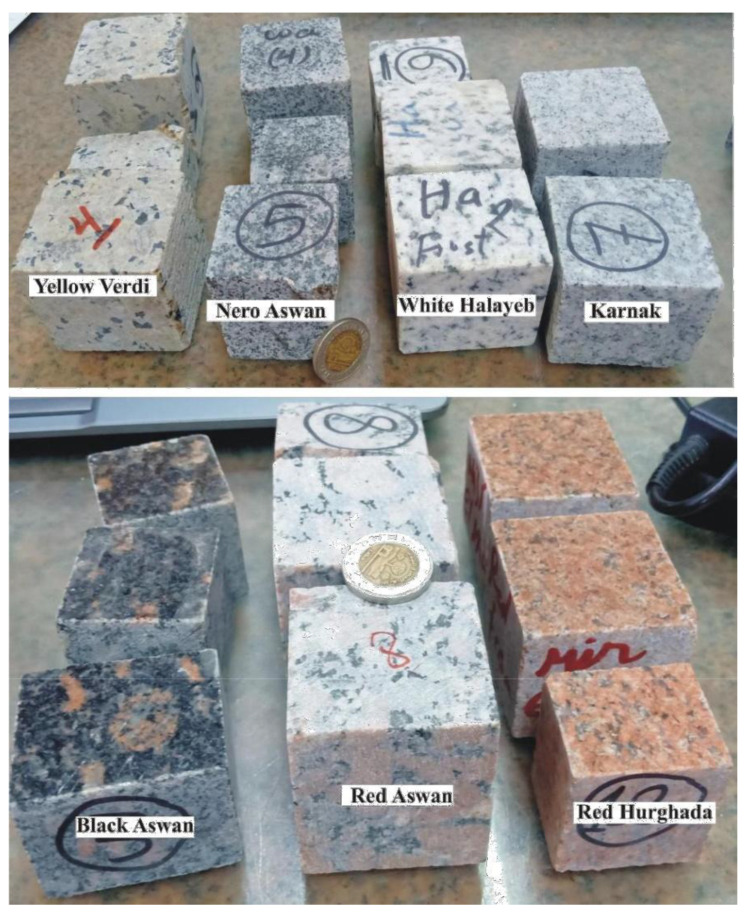
Photographs of the examined granitic rocks.

**Figure 3 materials-14-07290-f003:**
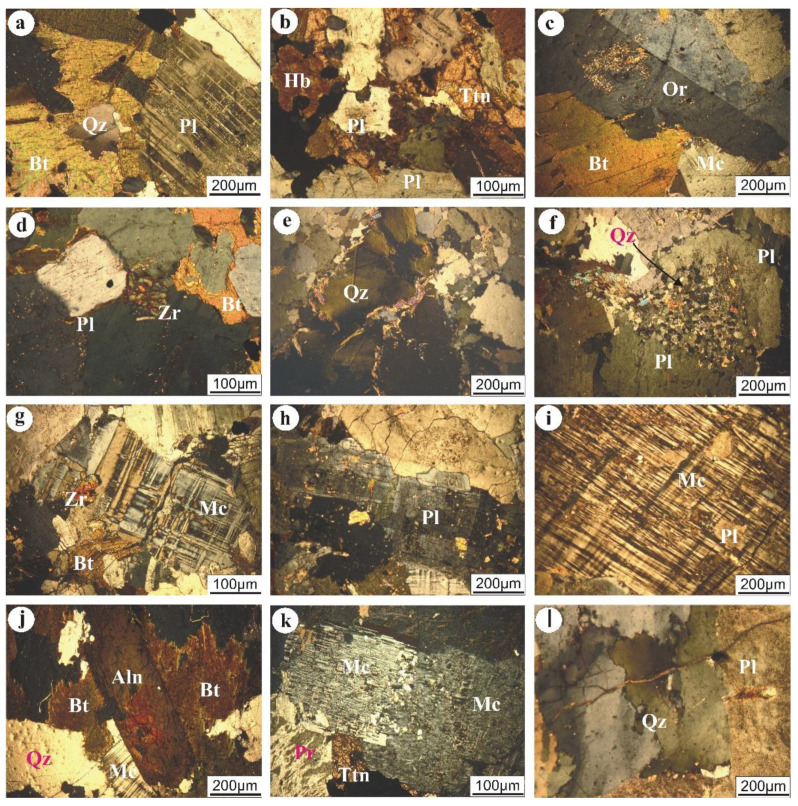
Photomicrographs of the studied magmatic rocks: black Aswan: (**a**) partially to completely chloritized biotite corroded by undulose quartz and plagioclase; (**b**) aggregation of titanite associated with plagioclase and titanite. Nero Aswan: (**c**) slightly sericitized orthoclase perthite corroded microcline and biotite; (**d**) well developed, euhedral zircon embedded in slightly saussuritized plagioclase. White Halayeb: (**e**) fine-to medium- grained undulose quartz fractured and filled by sericite; (**f**) zoned plagioclase engulfing fine-grained quartz. Karnak: (**g**) euhedral zircon enclosed in microcline, twisted and fractured biotite transformed to chlorite; (**h**) saussuritized and epidotized pericline plagioclase. Red Aswan: **i)** very coarse-grained, pristine, patchy microcline engulfing kaolinitized plagioclase; **j)** coarse-grained allanite surrounded by biotite crystals. Verdi: (**k**) titanite crystals enclosed by iron oxide, perthite and microcline perthite. Hurghada: (**l**) extended fracture filled by sericite intersecting extensive saussuritized plagioclase and undulose quartz.

**Figure 4 materials-14-07290-f004:**
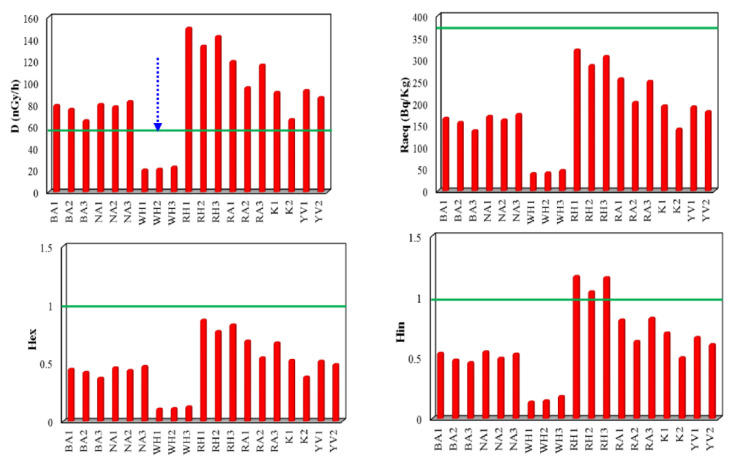
Absorbed dose rate (D), radium equivalent activity (Ra_eq_), external (H_ex_) and internal (H_in_) hazards of the examined samples.

**Table 1 materials-14-07290-t001:** Activity concentrations of ^226^Ra, ^232^Th, ^40^K (Bq/kg) and ^232^Th/^226^Ra ratio for the examined rocks.

Granite/Statistics Parameters	^226^Ra	^232^Th	^40^K	^232^Th/^226^Ra
Black Aswan				
Mean	29.60	44.44	803.37	1.56
SD	6.41	4.04	160.62	0.40
Min	22.20	40.40	626.00	1.21
Max	33.30	48.48	939.00	2.00
Nero Aswan				
Mean	25.90	55.21	855.53	2.20
SD	6.41	6.17	95.62	0.52
Min	22.20	48.48	751.20	1.70
Max	33.30	60.60	939.00	2.73
White Halayb				
Mean	15.17	4.71	292.13	0.32
SD	5.25	1.17	41.68	0.04
Min	11.10	4.04	244.14	0.29
Max	21.09	6.06	319.26	0.36
Red Hurghada				
Mean	111.00	86.19	939.00	0.78
SD	11.10	6.17	93.90	0.08
Min	99.90	80.80	845.10	0.69
Max	122.10	92.92	1032.90	0.84
Red Aswan				
Mean	44.40	92.92	1042.29	1.92
SD	11.10	12.99	74.22	0.28
Min	33.30	68.68	898.31	1.60
Max	55.50	92.92	1042.29	2.09
Karnak				
Mean	55.50	46.46	616.61	0.85
SD	15.70	8.57	123.94	0.09
Min	44.40	40.40	528.97	0.79
Max	66.60	52.52	704.25	0.91
Verdi				
Mean	49.95	44.44	968.74	0.91
SD	7.85	5.71	104.02	0.26
Min	44.40	40.40	895.18	0.73
Max	55.50	48.48	1042.29	1.09
Worldwide (UNSCEAR, 2010)	32.00	45.00	412.00	

**Table 2 materials-14-07290-t002:** Comparison of ^226^Ra, ^232^Th and ^40^K activity concentration in different areas.

Country	^226^Ra	^232^Th	^40^K	References
Egypt	103–2047	12.4–101.2	831.6–1394.6	[31]
Egypt	165–27,851	71–274	1048–1230	[32]
Egypt	12.4–534.4	56.6–169.8	398–1113	[33]
Egypt	137	82	1082	[34]
Nigeria	63.29	226.67	832.59	[35]
Saudi Arabia	28.82	34.83	665.08	[36]
Saudi Arabia	11	22	641	[37]
Palestine	71	82	780	[38]
Oman	17	18	379	[39]
Iran	77.4	44.5	1017.2	[40]
Jordan	41.52	58.42	897	[41]

## Data Availability

Data sharing is not applicable for this article.

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
