# Peer review of "Radiological Hazard Evaluation of Some Egyptian Magmatic Rocks Used as Ornamental Stone: Petrography and Natural Radioactivity"

_materials, 2021, doi:10.3390/ma14237290_

Round 1
Reviewer 1 Report
I have found the manuscript to be considerably improved during the revision. The revision is far from perfect, but in view of the effort put into the research and the revision and apart for some minor corrections concerning the English language, I see no more objections to ask the editor to accept the manuscript for publishing.
Author Response
Thanks for your recommendations.
The English Language was edited in the manuscript.
Reviewer 2 Report
I have modified some sentence in your manuscript to make them more readable, but further discussion may be more valuable for the manuscript. Some suggestions that I think they would be benefit for you are proposed as follow:
1 In the introduction chapter
how U and Th are distributed in the felsic rocks?
U and Th are incompatible elements, which often tend to be enriched in the upper continental crust and/or felsic intrusive rocks.
In the felsic rocks, Th and U mainly dwell in various refractory accessory phases, such as apatite, allanite, xenotime, monazite, titanite, thorite and zircon.
please refer Bea,1996; Jaupart et al., 2014.
- in the Results and discussion chapter
why the white Halyeb rocks have the low activity concentrations? what is the link with the lithology?
- in the Results and discussion chapter
Which one index is the most important for the hazard assessment? Including activity concentrations.
Radon, as the decay product of U and Th, has a short half-life (~3.8d). Whether it has health risks when high concentration in the closed room?
Whether the rocks with high U and Th contents will have health risks? and they can't be used as building materials? Such as indoor and outdoor.
What is the rock type that can be usually used as construction materials?

Author Response
1 In the introduction chapter
How U and Th are distributed in the felsic rocks?
Response: Protracted fractional crystallization of the magma yielding an enrichment of some elements such as Rb, Nb, Ta, U, Th, Zr and REEs. The previous elements abundance increase with increasing alkalinity (K2O+Na2O), (El Mezayen et al., 2019). In addition, Zircon, allanite, xenotime and thorite minerals are the main accessory minerals in felsic rocks (granitic rocks) and U and Th are more easily able to enter the lattice structures of these elements (e.g., Gromet & Silver,1983; Xiang et al., 2011). On the other hand, U and Th in felsic rocks as a result of hydrothermal activities that led to different alteration processes (El Mezayen et al., 2017, 2019).
U and Th are incompatible elements, which often tend to be enriched in the upper continental crust and/or felsic intrusive rocks.
Response: The dominant rock units of the upper continental crust is felsic rocks that are enriched with U and Th due to magmatic and post magmatic processes (hydrothermal) as discussed above.
In the felsic rocks, Th and U mainly dwell in various refractory accessory phases, such as apatite, allanite, xenotime, monazite, titanite, thorite and zircon.
Response: U and Th are more easily able to enter the lattice structures of these elements (e.g., Xiang et al., 2011).
Please refer Bea,1996; Jaupart et al., 2014.
Response: done.
Bea, F. Residence of REE, Y5 Th and U in Granites and Grustal Protoliths; Implications for the Chemistry of Crustal Melts. 1996. J. Petrol., 37, 521–552. https://doi.org/10.1093/petrology/37.3.521
Jaupart, C., Mareschal, J.C., Bouquerel, H., Phaneuf, C. The building and stabilization of an Archean Craton in the Superior Province, Canada, from a heat flow perspective. J. Geophys. Res. Solid Earth, 2014, 119, doi:10.1002/2014JB011018.2014
Petrographic investigation
Rapakivi textures are abundant. Quartz (22 vol.%) occurs as an anhedral crystal, fine- to medium-grained, that reveals undulose extinction.
Response: Black Aswan is characterized by the abundance of rapakivi textures [K-feldspar (pink crystal) crystals are mantled by plagioclase (white crystals)] in hand specimen. As a result of deformation processes, quartz crystals show undulose (wavy) extinction under a polarizing microscope.
- in the Results and discussion chapter
Why the white Halyeb rocks have the low activity concentrations? What is the link with the lithology?
Response: excellent question. White Halyeb is classified as tonalite (older granites), representing the oldest units in the current study. There is a strong relationship between the major mineralogical composition and accessory minerals such as zircon, allanite (Gromet & Silver, 1983). It consists mainly of plagioclase, quartz, biotite, and a minor amount of muscovite and accessory minerals. Lack of such minerals (e.g., Allanite, monazite, thorite, and zircon) led to low contents of activity concentrations. Therefore become more suitable as a decorative stone (ornamental).
- in the Results and discussion chapter
Which one index is the most important for the hazard assessment? Including activity concentrations.
Response: All indices are very important. However, the radiological parameters are applied to detect the radiation exposure for the public; the annual effective dose is the most parameter that is computed to detect the radiation dose. Based on the estimated value of AED can be identified the granites will be applied in building materials.
Radon, as the decay product of U and Th, has a short half-life (~3.8d). Whether it has health risks when high concentration in the closed room?
Response: This depends on the duration of the closed room. Therefore, it can be recommended to open the room for a few minutes to refresh the air inside the room.
Whether the rocks with high U and Th contents will have health risks? and they can't be used as building materials? Such as indoor and outdoor.
Response: Exposure to high dose with along time can affect the unfavourable health impacts as the deterioration of tissues and deoxyribonucleic acid (DNA) in genes and cancer and cardiovascular disease
What is the rock type that can be usually used as construction materials?
Response: Based on the current study, White Halyeb is the best granite that can be utilized in building materials.
Round 2
Reviewer 2 Report
- The second paragraph need to be polished,
Why the felsic granites have high U and Th contents? may depend on the U and/or Th-bearing accessary minerals, rather than primary uranium minerals. Commonly, Primary uranium minerals in the felsic rocks are accessory phases, not major minerals.
- What is the link between Petrography and Natural Radioactivity? A further discussion in the text may be more valuable for the manuscript, rather than reply to me.
Author Response
1. The second paragraph need to be polished,
Response: The paragraph has been polished.
Why the felsic granites have high U and Th contents? may depend on the U and/or Th-bearing accessary minerals, rather than primary uranium minerals. Commonly, Primary uranium minerals in the felsic rocks are accessory phases, not major minerals.
Response: The enrichment of U and Th in felsic rocks may be ascribed to magmatic segregation and hydrothermal activities that led to different alteration processes.
- What is the link between Petrography and Natural Radioactivity? A further discussion in the text may be more valuable for the manuscript, rather than reply to me.
Response: Thanks for your recommendation, which led to improving the manuscript. The link between and natural radioactivity has been discussed in the manuscript.
This manuscript is a resubmission of an earlier submission. The following is a list of the peer review reports and author responses from that submission.
Round 1
Reviewer 1 Report
The manuscript certainly has some merits in reporting radionuclide activities in magmatic rocks from Egyptian in order to evaluate these for their usefulnes as ornamental material. But going through the manuscript I have found a number of omissions and incorrect statements which have to be intensively addressed before considering a potential publication. Additionally the usage of English has to be dramatically improved.
I have given a large number of comments, corrections, suggestions directly in the manuscript which hopefully are helpful when revising the manuscript.

Author Response
Thank you for your questions and recommendations that improved the manuscript. The authors hope they answered all questions. Please find attached the submission of the carefully revised version of the manuscript in Ref., following the major comments and modification of the Reviewer.

Reviewer 2 Report
Saussurite is amineral aggregate.
Author Response
Thank you for your question that improved the manuscript. Please find attached the submission of the carefully revised version of the manuscript.

Reviewer 3 Report
1 Why the authors use the NaI γ-ray spectrometer? The throsy for the application of the detector?
2 Fig.4 (a), the D is higher than the standard, and the the Fig.4 (d).
3 How about the background of the testing experiment?
4 Why the stones will emit the rays? Or which rays will emit from U, Th...
5 I donot find any the discussion for the paper, i find the data are collected thorugh the testing results?
6 How about the novelty of the paper
7 Revise the lanaguage and the format.
Author Response
1 Why the authors use the NaI γ-ray spectrometer? The throsy for the application of the detector?
The NaI is one of the spectrometers which can use to detect the radionuclides activity concentrations. The essential features are the efficiency, linear energy response, size and shape flexibility, and low cost of NaI(Tl) detectors.
2 Fig.4 (a), the D is higher than the standard, and the the Fig.4 (d).
It is correct. The D and Hin are higher than the recommended limit of 59 nGy/h and 1, respectively, except for the values of White Halayeb.
3 How about the background of the testing experiment?
The background values are detected. The following text added in the manuscript “The background is reduced where the detector was enclosed in a cylindrical lead shield with a diameter of 15.7 cm, a length of 20.5 cm, and a thickness of 3.7 cm, with an attenuation factor of 0.16 (stopping around 84% of input photons) for 2.6 MeV gamma-rays.”
4 Why the stones will emit the rays? Or which rays will emit from U, Th...
The investigated study focused on the gamma emitted from the granites, including the uranium and thorium series and the radioactive potassium. Thus, radiological hazard parameters are estimated build on the radioactive concentration of uranium, thorium and potassium.
5 I donot find any the discussion for the paper, i find the data are collected thorugh the testing results?
The discussion is recorrected in the manuscript.
6 How about the novelty of the paper
The novelty is rewritten and described in the manuscript.
7 Revise the lanaguage and the format.
The language was revised in the manuscript.
Round 2
Reviewer 1 Report
I am deeply disappointed by the revision. The authors have largely failed to improve the manuscript and have not addressed a number of points which have been raised in my first review. In my opinion the manuscript is still far away of being acceptable for publication.
I have made a large number of additional comments in the authors cover letter.

Reviewer 3 Report
Agree